# Assessment of Ecosystem Service Value in Response to LULC Changes Using Geospatial Techniques: A Case Study in the Merbil Wetland of the Brahmaputra Valley, Assam, India

Durlov Lahon [1], Dhrubajyoti Sahariah [1], Jatan Debnath [1], Nityaranjan Nath [1], Gowhar Meraj [2,*], Pankaj Kumar [3], Shizuka Hashimoto [4] and Majid Farooq [2]

1   Department of Geography, Gauhati University, Guwahati 781014, India
2   Department of Ecology, Environment and Remote Sensing, Government of Jammu and Kashmir, Srinagar 190018, India
3   Institute for Global Environmental Strategies, Hayama 240-0115, Japan
4   Graduate School of Agricultural and Life Sciences, The University of Tokyo, Tokyo 113-8657, Japan
*   Correspondence: gowhar.60842@mygyanvihar.com or gowharmeraj@gmail.com

**Abstract:** The alteration of land use and land cover caused by human activities on a global scale has had a notable impact on ecosystem services at regional and global levels, which are crucial for the survival and welfare of human beings. Merbil, a small freshwater wetland located in the Brahmaputra basin in Assam, India, is not exempt from this phenomenon. In the present study, we have estimated and shown a spatio-temporal variation of ecosystem service values in response to land use and land cover alteration for the years 1990, 2000, 2010, and 2021, and predicted the same for 2030 and 2040. Supervised classification and the CA-Markov model were used in this study for land-use and land-cover classification and future projection, respectively. The result showed a significant increase in built-up areas, agricultural land, and aquatic plants and a decrease in open water and vegetation during 1990–2040. The study area experienced a substantial rise in ecosystem service values during the observed period (1990–2021) due to the rapid expansion of built-up areas and agricultural and aquatic land. Although the rise of built-up and agricultural land is economically profitable and has increased the study site's overall ecosystem service values, decreasing the area under open water and vegetation cover may have led to an ecological imbalance in the study site. Hence, we suggest that protecting the natural ecosystem should be a priority in future land-use planning. The study will aid in developing natural resource sustainability management plans and provide useful guidelines for preserving the local ecological balance in small wetlands over the short to medium term.

**Keywords:** ecosystem service values; ecology; wetland; land use/land cover; geo-spatial; Brahmaputra flood plain

## 1. Introduction

The benefits that people receive from ecosystems are referred to as ecosystem services (ES) [1,2]. A variety of essential ecosystem services for human well-being are provided by the global ecosystem, including regulating, provisioning, supporting, and cultural services [1,3], yet they are under increasing pressure due to the expansion, economic growth, and rising demand for natural resources by the human population [4]. Although there are many aspects to the value of ecosystems to human society, quantifying ecosystem service value (ESV) in monetary terms is a crucial tool for decision-makers to use when allocating resources rationally and also helps them understand user interest and the relative value of current ecological services [2,5–7]. Hence, significant effort has been put into estimating the ecosystem services' monetary value. Since the first quantification of the global ESV by Costanza et al. in 1997 [1], ESV estimation has been frequently employed as the basis of ecological benefit assessment that helps increase awareness and sustainable management

of ecosystems in a more precise method [8]. The average global ESV was estimated by Costanza et al. in 1997 to be USD 33 trillion/year in 1995. However, they modified their estimate in 2014 based on estimates of changes in land use and updated value of ES between 1997 and 2011. According to its updated valuation, the total annual worth of the world's ecosystem services reached USD 125 trillion in 2011 [9]. After this remarkable work, the volume of works on ESV estimation has grown, and many academicians and researchers employed Costanza et al.'s technique (both 1997 and 2014) to quantify the ESV of different land-use classes such as forest, grassland, agricultural land, water bodies, and wetland ecosystems. The conservation, management, and enhancement of the existing ecosystem services are necessary for practical advancements in society and economics. The functions and structure of ecosystems have changed over the last century and more as a consequence of changes in land use and land cover (LULC), environmental deterioration, global climate change, and associated resource depletion, which have affected the availability of ecosystem services [10,11]. Previous research has shown that the decline in carbon sequestration, loss of biodiversity, deterioration in water quality, and land degradation caused by cropland conversion, urbanization, and deforestation have significantly decreased ESV at local and regional levels [2,12]. Several studies have been conducted to evaluate the consequence of LULC alteration on ecosystem services worldwide using the coefficients given by Costanza et al. [1,9].

One of the most important natural resources, land, is constantly changing due to both natural and anthropogenic causes. The increasing human population and international economic development have triggered sudden changes in Earth's land surface over the past few decades [13]. Wetlands are among the most biologically diverse and productive ecosystems on the planet, and they provide a range of important ecosystem services. However, inland wetland ecosystems have undergone significant losses in the area and ecological function and are now facing global challenges due to the growing human population and the effects of global climate change. According to an estimate, the global total wetland area is a minimum of 1.5 to 1.6 billion hectares [14], but its area continues to decline, with conversion and loss ongoing in all parts of the world. LULC changes in wetlands have led to the loss and degradation of these important ecosystems, resulting in reduced water quality, increased vulnerability to floods, and a decline in biodiversity. Furthermore, the conversion of wetlands to other land uses can have significant impacts on local communities, which often rely on these ecosystems for their livelihood. Alternation of LULC and its spatial consequences on the wetland ecosystems are the major concern of geographers and environmentalists.

In the present study, the evaluation and prediction of Ecosystem Service Value (ESV) based on Land Use and Land Cover (LULC) changes are conducted using Remote Sensing (RS) and Geographic Information Systems (GIS) techniques due to their effectiveness in identifying and differentiating such alterations [11]. In recent years, satellite RS and GIS have become widespread technologies for mapping, quantifying, and detecting LULC change patterns due to their precise geo-referencing procedures, digital data processing technique, and repetitive data collection [15,16]. With the improvement of modern RS techniques and geo-analytic models, detecting the status and changing LULC using open-access RS data has become one of the cost-effective and reliable methods [17,18]. The up-to-date land cover data supplied by remote sensing can be used to evaluate past and recent LULC change trends and efficiently model future trends [19]. Thus, it has been widely considered an effective and powerful tool for natural resources management. In recent years, several models have been created by integrating remote sensing and GIS to project future LULC states efficiently, such as the Markov chain (MC) model, cellular automata (CA) model, logistic regression (LR) model, conversion of land use and its effects (CLUE) model, a modified cellular automata (SLEUTH) model, and an artificial neural network (ANN) model [19]. Among them, a widely used model for predicting LULC is the Cellular Automata-Markov (CA-Markov) model. It is a mixed model which includes

the concepts of both CA and the Markov chain [20]. In contrast to previous models, this combination of the CA and Markov models enables more accurate simulation [21].

The current research aims to assess the changing pattern of ESV in a floodplain wetland of the Brahmaputra River and its surrounding area with the alteration of LULC. The Brahmaputra, one of the world's largest rivers, passes through Bangladesh, India, Bhutan, and Tibet (China). The Brahmaputra River basin covers 70,634 sq. km in Assam out of a total of 194,413 sq. km in India (Water Resources, Govt. of Assam, 2019). The Assam valley, so named because it contains more than half the state's total land area, is formed by this river. The valley is around 700 km, extending from east to west, and varies in width between 65 and 100 km [22]. The valley is peppered with scattered knolls and marshy lowlands that are prone to annual flooding. It is also home to an oxbow lake, in addition to numerous large marshy regions and other wetlands. In terms of geomorphic, hydrologic, and biological importance, this valley's floodplain is, without question, its most important and productive ecosystem. However, industrialization and human encroachment have made these floodplains highly vulnerable. Humans have altered floodplains through a variety of land-use changes caused by urbanization, agriculture, industry, and mining. Overexploitation of natural resources, such as water, forests, land, etc., is a direct result of the region's growing population and rising human demands. Rapid changes in ESV are influencing LULC in the valley because of increasing urbanization, improved infrastructure, and extensive farming.

Research has frequently attempted to estimate the change of ESV owing to LULC modification from previous to current scenarios since it has a substantial impact on the alteration of ESV [7,23]. However, only a small number of researchers have investigated the possible changes in future valuations of ES related to the probable LULC changes. This has become very important due to anthropogenic climate change, rapid urbanization, and population growth which can put significant pressure on natural ecosystems [11,24]. As a result, the current study attempts to project LULC changes and landscape components for small floodplain wetlands and evaluate present and potential future natural capital and ESV. This is crucial for developing sustainable management plans that can anticipate such changes and implement policy frameworks that ensure ESV availability and improvements while maintaining the local ecological balance in the short and medium term.

## 2. Study Area

Assam's Brahmaputra valley is home to several large marshy lands, oxbow lakes, and various wetlands. One of them, Merbil, is a tiny freshwater lake located in the upper region of Brahmaputra valley, which falls in the Northeast India biogeographic zone. Its shape looks like an oxbow lake formed by the meandering of the Burhi Dihing River, the south-bank tributary of the river Brahmaputra. Its geographical position is 95°11′00″ E to 95°13′00″ E and 27°17′30″ N to 27°20′00″ N, respectively (Figure 1). The Charaideo district in the south bounds this lake, the Dihing Patkai National Park in the east, and the Burhi Dihing River in the north and west. As per Koppen's classification system, the region's climate can be classified as subtropical humid types controlled by different relief features and monsoonal circulation of the region [25]. The average annual temperature of the study area is 23.5 °C, and annual rainfall is 3034 mm. Rainfall is the most dominant climatic component of the study site, which starts in May and continues up to October. Although the main monsoonal rains start in May, the pre-monsoon showers begin mid-April [26]. The study site exhibits a topographically flat terrain from a geomorphologic standpoint, despite its close proximity of approximately 50 km to the Arakan Mountain's Patkai Range. In addition to surface and subsurface flow, precipitation is the main source of water for the wetland. Its water quality varies with the changes in seasons. The soil of the study area is sandy to clayey loamy, greyish in color, highly fertile, and formed as a result of the fluvial process of the Burhi Dihing River. Because of its unique climatic features and constantly changing weather, this freshwater lake provides shelter to various species of plants and animals. The wetland is also an important waterfowl habitat and supports

residential and migratory birds belonging to different families. In 2010, the 'Sasoni Merbil Eco-Tourism Project' started to preserve the biodiversity of the wetland. Various aquatic plants, including free-floating, floating-leaved, emergent, and submerged plants, are grown in the wetland. Over the last few decades, there has been a progressive increase in human settlement and involvement near the lake, which has resulted in a loss of species diversity and ecosystem balance [27]. Due to the growing human population at the study site, the ecological services provided by this lake and the surrounding environment have changed. Hence, Merbil Lake and its nearby area (500 m from the lake) have been selected as a study site to evaluate ESV with changes in LULC.

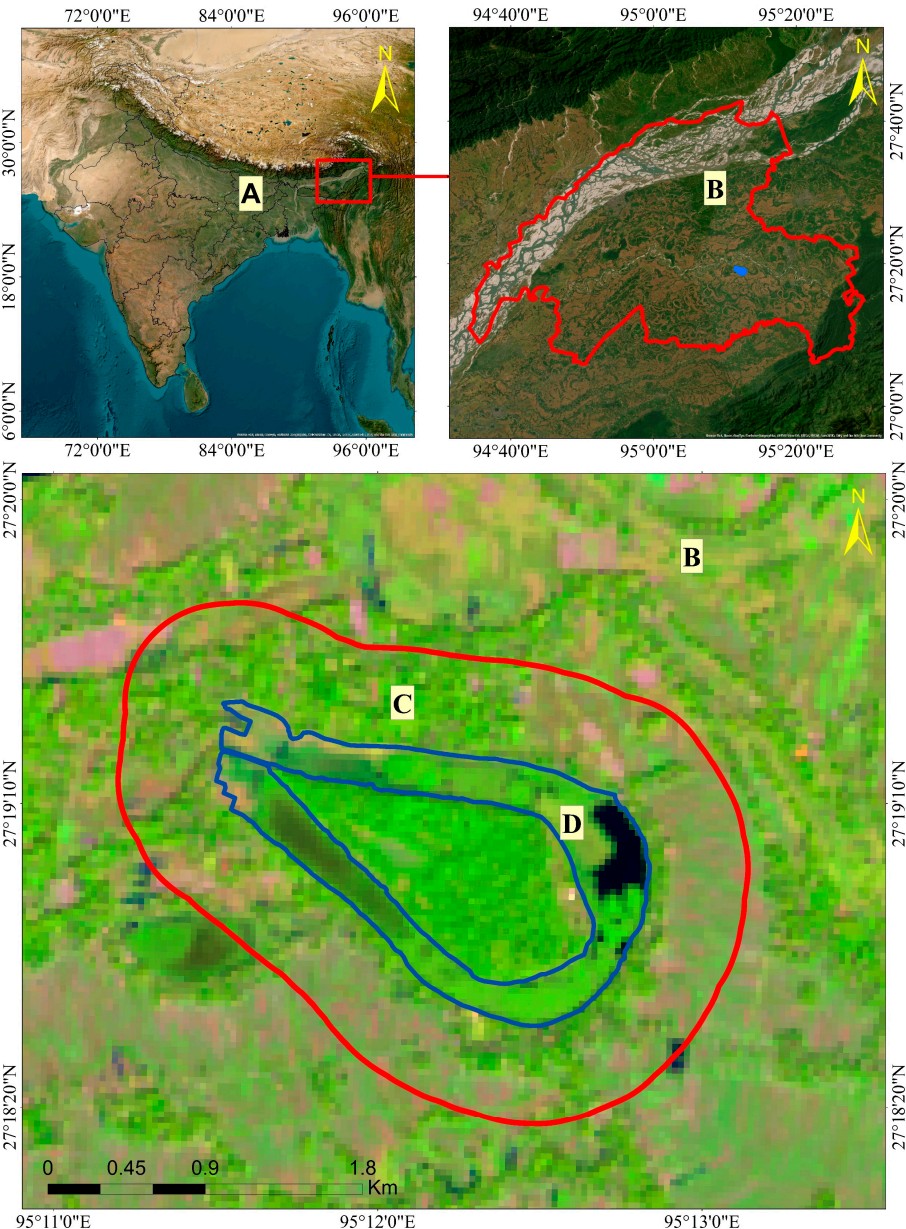

**Figure 1.** The map illustrates the location of the Merbil wetland within the broader geographical context. The red line encircles a 500-meter buffer zone surrounding the wetland. Key features include: (**A**) marking the boundary of India, (**B**) denoting the Dibrugarh district, (**C**) signifying the 500-meter buffer zone from the wetland, and (**D**) identifying the wetland itself. This visualization highlights the spatial relationship between the wetland and its surrounding areas, emphasizing the importance of understanding the potential impacts of land use and environmental changes on the wetland ecosystem.

## 3. Materials and Methods

### 3.1. Database

The extent of the different land-use types significantly influences the spatial distribution of ecosystem services. To assess the LULC, four Landsat images for the years 1990, 2000, 2010, and 2021 were taken from the United State Geological Survey (USGS) Earth Explorer (https://earthexplorer.usgs.gov; accessed on 31 March 2022). Landsat TM was used for the years 1990, 2000, and 2010, and Landsat OLI for the year 2021 (Path and Row/134-41). In order to obtain cloud-free images, the datasets were collected in the post-monsoon seasons. All Landsat images with a 30 m spatial resolution were used.

### 3.2. LULC Classification and Change Detection

In this study, five LULC types were considered, namely open water, aquatic plant, vegetation, agricultural land, and built-up areas for the years 1990, 2000, 2010, 2021, 2030, and 2040. Supervised classification was used for image classification, with MLC being the primary method used to produce LULC maps of the study area pixel by pixel [17]. Table 1 provides a breakdown of the various LULC categories. The kappa coefficient and overall accuracy statistics were used to evaluate the LULC classification portion of this study from 1990 to 2021. The validation procedure was aided by the use of satellite maps, Google Earth imagery, and GPS coordinates. Accuracy in classification was determined using a confusion matrix, the table of which was utilized to determine kappa accuracy, producer accuracy, user accuracy, and overall accuracy. Kappa and overall accuracy for all LULC-classified maps were consistently over 85%, making them suitable for further study.

**Table 1.** Description of LULC classes.

| Sl. No. | Class | Description |
|:---:|:---:|:---:|
| 1 | Open water | Clearwater surface (no vegetation cover) of the wetland |
| 2 | Aquatic plant | Water with vegetated cover, including free-floating, floating leaves, and emergent aquatic plants |
| 3 | Vegetation | Sparsely vegetated areas, densely vegetated areas, and grassland |
| 4 | Agricultural land | Cropland, plantation cultivated land, and agricultural fallow land |
| 5 | Built-up area | Settlements, road networks, concrete surfaces, and commercial buildings |

The change detection analysis shows the conversions from one type of LULC class to another during a specific period [13]. The temporal nature of LULC change was monitored and examined using the LULC change detection map. A post-classification strategy for change detection was used to evaluate the modifications. In this research, the change detection was performed in ArcGIS 10.8 using the change matrix approach (raster polygon). The latter depicts the change in LULC for every period from 1990 to 2021. Further, the alteration of LULC for 2030 and 2040 was also predicted to better recognize the futuristic change in this area. The degree of change for each class was measured using Equation (1), and the percentage of change for each LULC class was measured using Equation (2)

$$C_i = L_i - B_i \tag{1}$$

$$P_i = \frac{L_i - B_i}{B_i} \times 100 \tag{2}$$

where $C_i$ represents the weightage of change in class '$i$'; $P_i$ represents the percentage of change in class '$i$'; $B_i$ represents the base image, and $L_i$ denotes the current date image.

### 3.3. LULC Change Prediction Using the CA-Markov Model

The CA-Markov model was used to project possible future scenarios of changing LULC [28,29]. This model combines the Markov chain and CA algorithms to forecast future LULC alteration trends. It provides policymakers with knowledge and guidance to assess and understand the processes influencing change in vegetation through the generation of LULC scenarios [30]. This is crucial for land-use management and policymaking [31]. Changes in future LULC in our study area were projected using the IDRISI TerrSet software. The stages taken throughout this process follow the creation of a transition probability image utilizing the base map, transition suitability of the image, and LULC maps from 1990, 2000, 2010, and 2021 images [32,33]. The Markovian Transition Estimator technique was then used to generate the transition probabilities for 2000 and 2010 to model the LULC for 2021 (for validation purposes), as well as for 2010 and 2021 to model the LULC maps for 2030 and 2040. The transition suitability was calculated using the constraints and variables of the MCA (multi-criteria evaluation) module [34,35]. Finally, the transition suitability map, base map, and transition probabilities map were used to simulate the 2030 and 2040 LULC.

The CA model is shown as follows [29,36].

$$S(t, t+1) = f(S(t), N) \tag{3}$$

where $S(t+1)$ is the status of the system at time $(t, t+1)$, and is the function by the state probability of given time ($N$)

This model is frequently used to monitor LULC, ecological modeling, and change simulation and predict the degree of land-use modification and the stability of future land development in the area of concern. The prediction of LULC changes is calculated using Equation (4).

$$S(t, t+1) = P_{ij} \times S(t), \tag{4}$$

where $S(t)$ is the status of system at time $t$, $S(t+1)$ is the status of the system at time $t + 1$; $P_{ij}$ is the transition probability matrix of the given state, and is evaluated as follows [17,37].

$$= ||P_{ij}|| = \left\| \begin{matrix} P1,1 \; P1,2 \; P1,N \\ P1,1 \; P1,2 \; P2,N \\ \ldots \ldots \ldots \\ PN,1 \; PN,2 \; PN,N \end{matrix} \right\|, \; (0 \le P_{ij} \le 1), \tag{5}$$

where $P_{ij}$ stands for the probability of transforming from present state $i$ to another state $j$ in succeeding time; $P$ denotes the transition probability; and $PN$ is the state probability of any time. The low transition shall have a probability of 0, whereas the high transition shall have a probability close to 1 [37]. The Markov chain evaluates precisely how much the land will transform from the previous year to the forecast year. Figure 2 shows the flowchart of the adopted methodology.

### 3.4. Assessment of Ecosystem Service Value (ESV)

To assess the ESV, the LULC of the study site was analyzed for the years 1990, 2000, 2010, and 2021 and predicted for the years 2030 and 2040. The LULC data for each relevant year was used as a proxy for the measurement of ESV, and the corresponding area in hectares was summarized in the GIS environment [38]. The principles and methods for the value assessment of ecosystem services were based on the coefficients provided by Costanza et al. (1997) and their updated coefficients (2014), which were assigned to each LULC class (Table 2). Costanza et al. estimated global ecosystem services in 1997 using the USD value in 1995, and in 2014 using USD value in 2007. The 1997 coefficient values were suitable for estimating ESV during that time period, while the 2014 coefficient values were more relevant for recent decades. To facilitate comparison of ESV across different time periods, the present study used both sets of coefficient values to estimate ESV. The total ESV

for individual LULC classes was calculated by multiplying the associated value coefficients by the total area of each LULC type in hectares. To compute the total ESV of the landscape for each reference year, the values for the LULC types in each year were summed. The coefficient values provided by Costanza et al. were selected for the present study because they offered estimates for 16 major biomes and 17 ecosystem services, making it the most comprehensive set of valuation coefficients currently available to us. Table 3 presents the specific coefficient values recommended by Costanza et al. (1997, 2014).

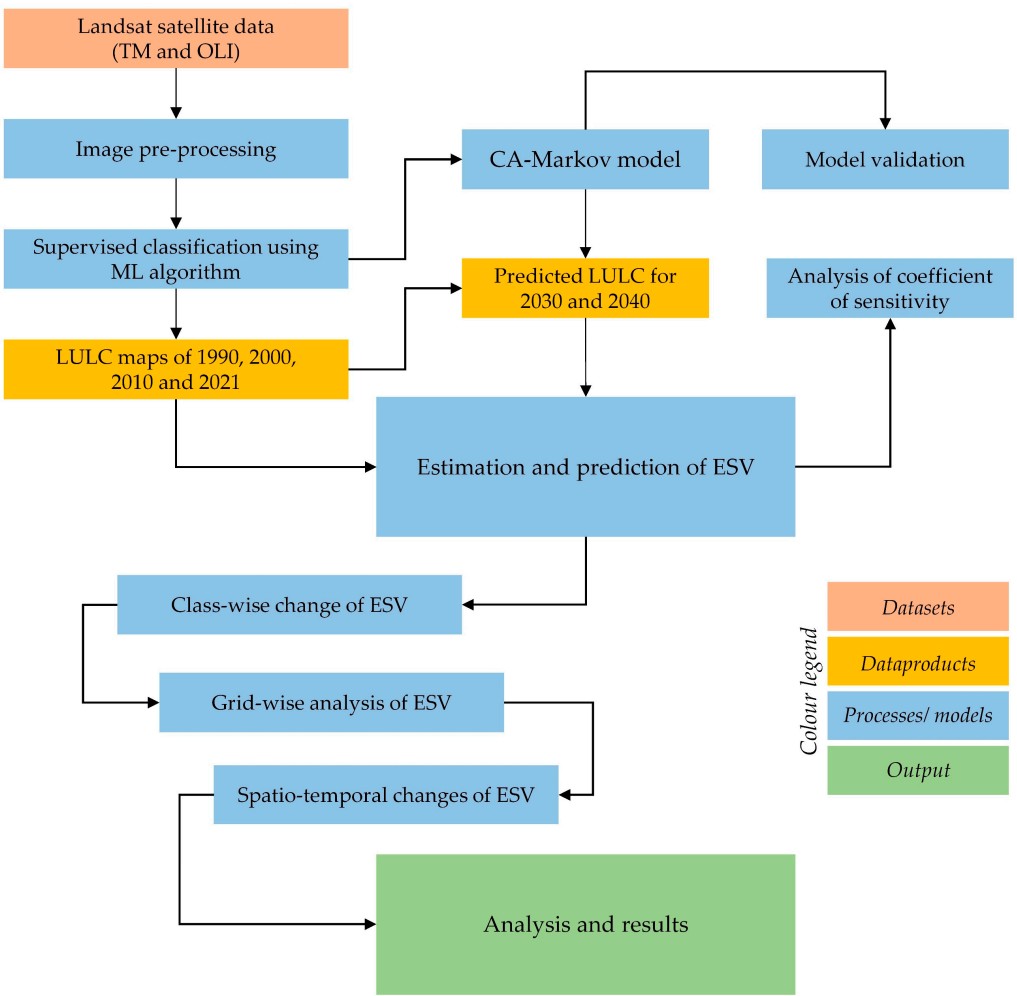

**Figure 2.** Methodology flowchart adopted in the study.

**Table 2.** Description of ecosystem service coefficient as per Costanza et al. (1997, 2014).

| LULC Types | | Coefficient Value of Ecosystem Services (USD ha$^{-1}$ yr$^{-1}$) | |
|---|---|---|---|
| | Similar Biome (Costanza et al., 2014) | Costanza et al. (1997) | Costanza et al. (2014) |
| Open water | Lake/River | 11,727 | 12,512 |
| Aquatic plant | Swamps | 27,021 | 25,681 |
| Vegetation | Forest | 1338 | 3800 |
| Agricultural land | Cropland | 126 | 5567 |
| Built-up area | Urban | 0 | 6661 |

**Table 3.** Ecosystem services' yearly value coefficients (in USD ha$^{-1}$ year$^{-1}$).

| S. No. | Ecosystem Service Types | Open Water | | Aquatic Plant | | Vegetation | | Agricultural Land | | Built-Up | |
|--------|-------------------------|------|------|------|------|------|------|------|------|------|------|
| | | 1997 | 2014 | 1997 | 2014 | 1997 | 2014 | 1997 | 2014 | 1997 | 2014 |
| *Provisioning* | | | | | | | | | | | |
| 1 | Food production | 57 | 106 | 65 | 614 | 59 | 270 | 75 | 2323 | - | - |
| 2 | Raw materials | - | - | 68 | 539 | 191 | 152 | - | 219 | - | - |
| 3 | Genetic resource | - | - | - | 99 | 22 | 448 | - | 1042 | - | - |
| 4 | Water supply | 2922 | 1808 | 10,488 | 408 | 4 | 143 | - | 400 | - | - |
| *Regulating* | | | | | | | | | | | |
| 5 | Gas regulation | - | - | 366 | - | - | 4 | - | - | - | - |
| 6 | Disturbance regulation | - | - | 9991 | 2986 | 3 | 19 | - | - | - | - |
| 7 | Erosion control | - | - | - | 2607 | 132 | 100 | - | 107 | - | - |
| 8 | Pollination | - | - | - | - | - | 9 | 19 | 22 | - | - |
| 9 | Climate regulation | - | - | - | 488 | 194 | 711 | - | 411 | - | 905 |
| 10 | Biological control | - | - | - | 948 | 3 | 169 | 32 | 33 | - | - |
| 11 | Water regulation | 7514 | 7514 | 41 | 5606 | 3 | 3 | - | - | - | 16 |
| 12 | Waste-treatment | 918 | 918 | 2289 | 3015 | 120 | 120 | - | 396 | - | - |
| *Supporting* | | | | | | | | | | | |
| 13 | Nutrient cycling | - | - | - | 1713 | 498 | 66 | - | - | - | - |
| 14 | Soil formation | - | - | - | - | 14 | 14 | - | 532 | - | - |
| 15 | Habitat/refugia | - | - | 605 | 2455 | - | 619 | - | - | - | - |
| *Cultural* | | | | | | | | | | | |
| 16 | Cultural | - | - | 2430 | 1992 | 3 | 1 | - | - | - | - |
| 17 | Recreation | 317 | 2166 | 678 | 2211 | 91 | 953 | - | 82 | - | 5740 |
| | Total ESV | 11,728 | 12,512 | 27,021 | 25,681 | 1338 | 3800 | 126 | 5567 | - | 6661 |

Costanza et al. gave coefficients for estimating the value of ecosystem services in 16 major biomes (Table 3). However, the LULC classes used in our study are not exactly the same as those in Costanza et al. study. Hence, we used those coefficient values of our land-use class that are similar to the biome they provided. To calculate the ESV of vegetation, agricultural land, and built-up areas in our study, we utilized the coefficient values for forest, cropland, and urban areas, respectively, from Costanza et al. study. This approach was taken because these biomes have similar characteristics to our study's LULC classes. The service value of open water was estimated using the coefficient value of lakes and rivers due to their similar nature. On the other hand, the ESV of the aquatic plant class was estimated using the coefficient of swamps because these plants are typically found in swampy areas, which are integral parts of wetland ecosystems and serve as habitats for a variety of flora and fauna [39].

To compute the ESV of LULC types, LULC functions, and total ESV, the following equations were applied:

$$ESV_k = \sum_f A_k \times VC_{kf} \tag{6}$$

$$ESV_f = \sum_k A_k \times VC_{kf} \tag{7}$$

$$ESV_t = \sum_k \sum_f A_k \times VC_{kf} \tag{8}$$

where $ESV_f$, $ESV_k$, and $ESV_t$ represent the ecosystem service value of LULC type $k$, LULC function $f$, and the total value of ecosystem services, respectively. $A_k$ denotes the area (ha) of the land-use type $k$, $VC_{kf}$ is the equivalent value coefficient (USD ha$^{-1}$ year$^{-1}$) of a specific land-use type '$k$' and ecosystem service '$f$', respectively. For evaluating the spatio-temporal changes of ESVs during different periods, the value coefficient of each ecosystem service and land-use category was estimated using 2007 unit values (USD ha$^{-1}$) [38,40,41].

The changing dynamic of ESVs was estimated by calculating the difference between the estimated values in each reference year using the following equation:

$$ESV_c = \frac{ESV_{end} - ESV_{start}}{ESV_{start}} \times \frac{1}{t} \times 100\% \tag{9}$$

where *ESVc* is the change rate of ESV, $ESV_{end}$, and $ESV_{start}$ refer to the estimated ESV at the start and the end of the study period, and *t* represents the period.

There are ambiguities in the ESV valuation since the biomes we selected as proxies for the LULC types are not always completely suited to Costanza et al. (1997) ESV model. As a result, sensitivity analysis was employed to determine how the ESV would respond to a change in the value coefficient [42]. Accordingly, each of the value coefficients for open water, aquatic plant, vegetation, agricultural land, and the built-up area was adjusted by 50 percent, and the relevant coefficient of sensitivity (CS) was computed using Equation (10) as in Kreuter et al. [43].

$$CS = \frac{(ESV_j - ESV_i)/ESV_i}{(VC_{jk} - VC_{ik})/VC_{ik}} \tag{10}$$

where *ESV* and *VC* indicate the measured ecosystem service value and the value coefficient, respectively. '*i*' represents the initial value, '*j*' represents the adjusted values, while '*k*' denotes the LU types. *CS* > 1 indicates the calculated ESV, which is elastic with respect to the coefficient, while *CS* < 1 indicates the measured ESV, which is inelastic, and the output will also be reliable if the value coefficient has relatively low accuracy [43,44].

The study site was divided into 200 × 200 m grids to estimate the spatial variation of ESV, and the ESV of various LULC classes in each grid was measured. Kriging was used to evaluate the spatial variation of ESV in the study site. The present study estimated the ESV for the years 1990, 2000, 2010, and 2021 based on the observed LULC maps and predicted the ESV for 2030 and 2040 based on the predicted LULC maps. In addition, ESV was classified into five categories based on grid-wise analysis: very low (USD 12,000), low (USD 12,000–16,000), moderate (USD 16,000–20,000), high (USD 20,000–24,000), and very high (>USD 24,000).

## 4. Results

### 4.1. Analysis of LULC Dynamics

The multi-temporal change of LULC was monitored and projected for the years 1990, 2000, 2010, 2030, and 2040 through five different classes: open water, aquatic plants, vegetation, agricultural land, and built-up area. Table 4 displays the area of various LULC types, and Figure 3 depicts the spatial distribution pattern of LULC and the pattern of future LULC. Vegetation is the most dominant land-use class of the study area and continued to decline throughout the study period. From 283 ha in 1990, it had decreased to 212.84 ha in 2021, and by 2040, it is expected to decline further to 155.56 ha. Another crucial land-use class at the study site is agricultural land, which grew from 187.11 ha in 1990 to 227.84 ha in 2021 and may reach 286.57 ha in 2040. The built-up area of the study site is gradually increasing. In 1990, the area occupied by this class was only 15.7 ha, but it grew to 27.3 ha, 36.22 ha, and 43.36 ha in 2000, 2010, and 2021, respectively. With the continual growth of built-up areas in the study site, agricultural land has expanded while vegetation has steadily diminished. Open water and aquatic plant are very dynamic land-use classes of the study area that are interrelated. The area of open water has reduced while the area of aquatic plants has expanded. In 1990, 2000, 2010, and 2021, the area occupied by aquatic plants was 74.98 ha, 83.3 ha, 66.14 ha, and 95.76 ha, respectively. It was also predicted that in the years 2030 and 2040, the area would be 99.23 ha and 97.18 ha. Open water has experienced a declining trend in all years except 2010. From 28.61 ha in 1990, it had decreased to 10.61 ha in 2021, and by 2040, it is expected to decline further to 1.35 ha.

**Table 4.** Areal extents of land-use/land-cover (LULC) types from 1990 to 2040.

| LULC Classes | 1990 Area (ha) | 2000 Area (ha) | 2010 Area (ha) | 2021 Area (ha) | 2030 Area (ha) | 2040 Area (ha) |
|---|---|---|---|---|---|---|
| Open water | 28.61 | 26.94 | 43.91 | 10.61 | 3.50 | 1.35 |
| Aquatic plant | 74.98 | 83.3 | 66.14 | 95.76 | 99.23 | 97.18 |
| Vegetation | 283.25 | 253.3 | 265.52 | 212.8 | 181.15 | 155.56 |
| Agricultural land | 187.11 | 199.07 | 178.01 | 227.84 | 259.50 | 286.57 |
| Built-up area | 15.7 | 27.3 | 36.22 | 43.36 | 45.27 | 48.04 |

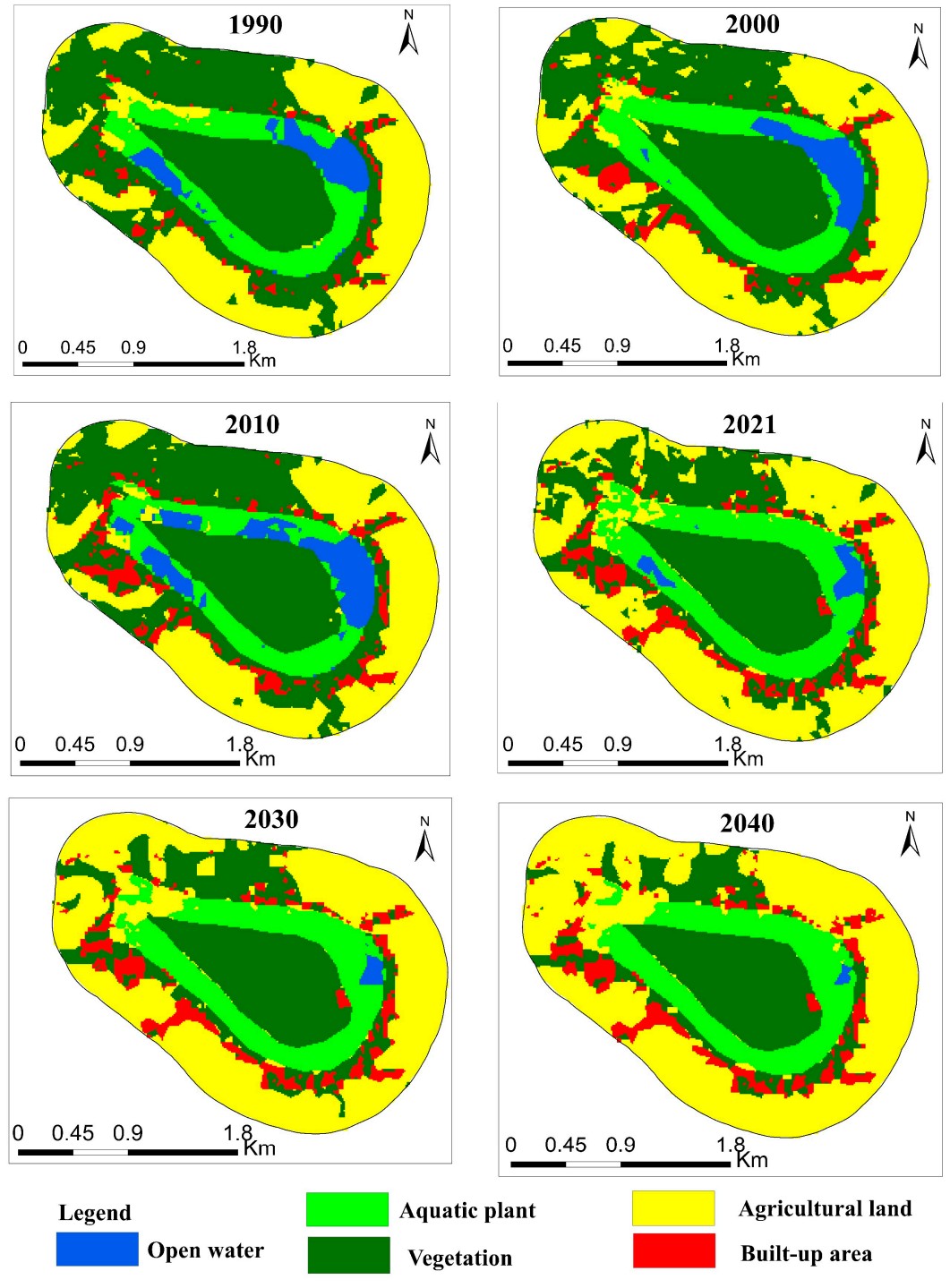

**Figure 3.** Land-use/land cover map of the study area for 1990, 2000, 2010, 2021, 2030, and 2040.

In Table 5, the data presented in the 2nd and 3rd columns, labeled as A and P, respectively, represent the percentage share of individual LULC classes used for model validation. In this study, the CA-Markov model was employed to predict future land-use and land cover (LULC) patterns in the study area. To assess the model's performance and its suitability for predicting LULC changes, a Chi-square test ($X^2$) was conducted to determine its goodness of fit using the Equation (11). The results of the Chi-square test were found to be equal to 5.59, which demonstrates a good fit between the model's predictions and the observed data for the study area. This indicates that the CA-Markov model is an appropriate tool for predicting future LULC patterns in the area under investigation.

$$X2 = \sum \frac{(P - A)^2}{A} = 5.59; \ df = 4 \ and \ X^2_{0.05}(4) = 9.49 \tag{11}$$

**Table 5.** Validation of change prediction based on actual (2021) and predicted (2021) LULC map using the Chi-square test ($X^2$).

| LULC Classes | Actual Area (A) (2021) | Predicted Area (P) (2021) | $(P-A)^2/A$ |
|---|---|---|---|
| Open water | 1.8 | 3.49 | 1.59 |
| Aquatic plant | 16.22 | 12.3 | 0.95 |
| Vegetation | 36.05 | 44.75 | 2.1 |
| Agricultural land | 38.59 | 33.62 | 0.64 |
| Built-up area | 7.34 | 5.84 | 0.31 |
| Total | 100 | 100 | 5.59 |

Additionally, the current study estimated the decadal change rate of relevant LULC types (Table 4). The area and percentage-wise changes in different LULC categories of five periods, i.e., 1990–2000, 2000–2010, 2010–2021, 2021–2030, and 2030–2040 were shown in Table 6. It was observed that between 1990 and 2000, agricultural land, built-up areas, and aquatic plant coverage increased by a total of 11.95 ha, 11.61 ha, and 8.32 ha, respectively. Concurrently, there was a decrease in vegetation and open water, with a total reduction of 29.95 ha and 1.67 ha, respectively. In the period 2000–2010, the area under open water, vegetation, and built-up increased by 16.96 ha, 12.23 ha, and 8.91 ha, respectively. In contrast, aquatic plant and agricultural land classes were reduced by 17.16 and 21.05 ha, respectively. The built-up area, along with the agricultural and aquatic plant, rose by 7.14 ha, 49.82 ha, and 29.62 ha, respectively, between 2010 and 2021. On the contrary, open water and vegetation areas dropped by 33.29 ha and 52.72 ha, respectively, in the same period. Agricultural land, aquatic plant, and built-up areas exhibit positive growth between 2021 and 2030, with respective areas of 31.66 ha, 3.47 ha, and 1.91 ha; while vegetation and open water areas decreased by 31.55 ha and 7.11 ha, respectively. From 2030–2040, the built-up and agricultural land again increased from previous periods, correspondingly with an area of 2.77 ha and 27.07 ha. This period shows a significant decrease in vegetation, with an area of 25.69 ha less than the previous period. Furthermore, the area under open water and the aquatic plant was reduced by 2.15 ha and 2.05 ha, respectively.

*4.2. Estimation and Prediction of Ecosystem Service Value (ESV)*

The value of ecosystem services offered by the LULC categories has been estimated and predicted for 1990, 2000, 2010, 2021, 2030, and 2040 (Table 7) using the coefficient value given by Costanza et al. (1997 and 2014). The aquatic plant type contributes the highest ESV of USD $2026.03 \times 10^3$, USD $2250.85 \times 10^3$, USD $1787.17 \times 10^3$, USD $2585.1 \times 10^3$, USD $2681.29 \times 10^3$, and USD $2625.9 \times 10^3$ as per the 1997 coefficient in 1990, 2000, 2010, 2021, 2030, and 2040, respectively. On the other hand, the ESV of the same class as per the 2014 coefficient was USD $1925.56 \times 10^3$, USD $2139.23 \times 10^3$, USD $1698.54 \times 10^3$, USD $2456.9 \times 10^3$, USD $2548.33 \times 10^3$, and USD $2495.68 \times 10^3$ in the years 1990, 2000, 2010, 2021, 2030, and 2040, respectively. After this, the vegetation contributed a significant amount of

ESV of USD 378.99 × 10³, USD 338.92 × 10³, USD 355.27 × 10³, USD 284.73 × 10³, USD 242.38 × 10³, and USD 208.14 × 10³, respectively, (as per the 1997 coefficient) in 1990, 2000, 2010, 2021, 2030, and 2040.

**Table 6.** Decadal changes of LULC in the study site between 1990 and 2040.

| LULC Class | 1990–2000 | | 2000–2010 | | 2010–2021 | | 2021–2030 | | 2030–2040 | | 1990–2040 | |
| --- | --- | --- | --- | --- | --- | --- | --- | --- | --- | --- | --- | --- |
| | Area in ha | Area in % | Area in ha | Area in % | Area in ha | Area in % | Area in ha | Area in % | Area in ha | Area in % | Area in ha | Area in % |
| Open water | −1.67 | −5.84 | 16.96 | 62.96 | −33.29 | −75.82 | −7.11 | −67.01 | −2.15 | −61.43 | −27.26 | −95.61 |
| Aquatic plant | 8.32 | 11.1 | −17.16 | −20.61 | 29.62 | 44.78 | 3.47 | 3.62 | −2.05 | −2.06 | 22.20 | 29.61 |
| Vegetation | −29.95 | −10.57 | 12.23 | 4.83 | −52.72 | −19.86 | −31.55 | −14.82 | −25.7 | −14.18 | −127.69 | −45.08 |
| Agricultural land | 11.95 | 6.39 | −21.05 | −10.58 | 49.82 | 27.99 | 31.66 | 13.90 | 27.07 | 10.43 | 99.46 | 53.15 |
| Built-up area | 11.61 | 73.94 | 8.91 | 32.65 | 7.14 | 19.72 | 1.91 | 4.40 | 2.77 | 6.11 | 32.34 | 205.98 |

**Table 7.** Various LULC types' provided ESV in 10³ USD from 1990 to 2040.

| LULC Classes | 1990 | | 2000 | | 2010 | | 2021 | | 2030 | | 2040 | |
| --- | --- | --- | --- | --- | --- | --- | --- | --- | --- | --- | --- | --- |
| | *Cc1997 | Cc2014 | Cc1997 | Cc2014 | Cc1997 | Cc2014 | Cc1997 | Cc2014 | Cc1997 | Cc2014 | Cc1997 | Cc2014 |
| Open water | 335.51 | 357.97 | 315.93 | 337.07 | 514.93 | 549.4 | 124.42 | 132.75 | 41.04 | 43.79 | 15.83 | 16.89 |
| Aquatic plant | 2026.03 | 1925.56 | 2250.85 | 2139.23 | 1787.17 | 1698.54 | 2587.53 | 2459.21 | 2681.29 | 2548.33 | 2625.9 | 2495.68 |
| Vegetation | 378.99 | 1076.35 | 338.92 | 962.54 | 355.27 | 1008.98 | 284.73 | 808.64 | 242.38 | 688.37 | 208.14 | 591.13 |
| Agricultural land | 23.58 | 1041.64 | 25.08 | 1108.22 | 22.43 | 990.98 | 28.71 | 1268.39 | 32.7 | 1444.64 | 36.1 | 1594.78 |
| Built-up area | 0 | 104.58 | 0 | 181.85 | 0 | 241.26 | 0 | 288.82 | 0 | 301.54 | 0 | 319.99 |
| Total | 2764.11 | 4506.10 | 2930.78 | 4728.91 | 2679.80 | 4489.16 | 3025.39 | 4957.93 | 2997.41 | 5026.67 | 2885.97 | 5018.47 |

*Cc represents Costanza et al. coefficients.

According to the coefficient of 2014, the ESV of vegetation was USD 1076.35 × 10³, USD 962.54 × 10³, USD 1008.98 × 10³, USD 808.64 × 10³, USD 688.37 × 10³, and USD 591.13 × 10³ in 1990, 2000, 2010, 2021, 2030, and 2040 correspondingly. Costanza et al. (2014) revised their coefficient value in 2014 and increased the value of vegetation and agricultural land. As a result, a notable difference has been observed between 1997 and 2014 for the ESV of the vegetation and agricultural land. In 1990, 2000, 2010, 2021, 2030, and 2040, the ES value of agricultural land according to the 1997 coefficient was USD 23.58 ×10³, USD 25.08 × 10³, USD 22.43 × 10³, USD 28.71 × 10³, USD 32.7 × 10³, and USD 36.1 × 10³, respectively, whereas the ESV according to the 2014 coefficient was USD 1041.58 × 10³, USD 1108.22 × 10³, USD 990.98 × 10³, USD 1268.39 × 10³, USD 1444.64 × 10³, and USD 1594.78 × 10³. Open water also contributed a significant amount of ESV in the study site. As per the 1997 coefficient, it contributes ESV of USD 335.51 × 10³, USD 315.93 × 10³, USD 514.93 × 10³, USD 124.42 × 10³, USD 41.04 × 10³, and USD 15.83 × 10³ in 1990, 2000, 2010, 2021, 2030, and 2040, respectively. On the contrary, as per the 2014 coefficient, it contributes to ESV, USD 357.97 × 10³, USD 337.07 × 10³, USD 549.4 × 10³, USD 132.75 × 10³, USD 43.79 × 10³, and USD 16.89 × 10³ in 1990, 2000, 2010, 2021, 2030, and 2040, respectively. Since the built-up area was not given a coefficient value by Costanza et al. (1997), the ESV of the built-up class would be zero each year based on the coefficient of that year. However, Costanza et al. revised their coefficient value in 2014 and generated an updated coefficient table where they assigned a value for the built-up area. According to the revised coefficient (2014), the ESV of the built-up area was USD 104.58 × 10³, USD 181.85 × 10³, USD 241.26 × 10³, USD 288.82 × 10³, USD 301.54 × 10³, and USD 319.99 × 10³ in 1990, 2000, 2010, 2021, 2030, and 2040, respectively.

The class-wise and total change of the ESV (in percent) from 1990–2021 (observed period) and 2021–2040 (predicted period) was computed based on Costanza et al. (1997 and 2014), as shown in Table 8. According to the 1997 coefficient, the rate of a total change of ESV in the observed periods (1990–2021) and predicted periods (2021–2040) was/would be 9.45 percent and −4.60 percent, respectively. On the other hand, using the 2014 coefficient, the total ESVs net change over the observed and predicted period was/would be 9.97 percent and 1.22 percent, respectively. In the observed period, the

ESV of open water and vegetation decreased by 62.91 percent and 24.87 percent (based on both 1997 and 2014 coefficients), respectively. In contrast, the ESV of aquatic plants, agricultural land, and built-up area increased by 27.71 percent and 21.75 percent (nil for the built-up) as per the coefficient of 1997, and 27.6 percent, 21.77 percent, and 176.17 percent as per the coefficient of 2014, respectively. In the predicted period (2021–2040), the ESV of open water and vegetation declined at a much higher rate compared to the observed period, by −87.28 percent and −26.89 percent, respectively, based on the 1997 coefficient and −87.27 percent and −26.89 percent, respectively, based on the 2014 coefficient given by Costanza et al. On the contrary, during the predicted period, the total ESV of agricultural land, aquatic plants, and built-up area showed positive change (both the 1997 and 2014 coefficients) (Table 8).

**Table 8.** The change rate of ESV during the observed period (1990–2021) and predicted period (2021–2040).

| LULC Classes | 1990–2021 | | 2021–2040 | |
|---|---|---|---|---|
| | *\*Cc*1997 | *Cc*2014 | *Cc*1997 | *Cc*2014 |
| | Change in % | | | |
| Open water | −62.91 | −62.91 | −87.28 | −87.27 |
| Aquatic plant | 27.71 | 27.6 | 1.57 | 1.48 |
| Vegetation | −24.87 | −24.87 | −26.9 | −26.89 |
| Agricultural land | 21.75 | 21.77 | 25.74 | 25.73 |
| Built-up area | 0 | 176.17 | 0 | 10.8 |
| Total change | 9.45 | 10.02 | −4.60 | 1.22 |

*\*Cc* represents Costanza et al. coefficients.

Table 9 shows the estimated values for different ecosystem functions of service categories (provisioning, regulating, supporting, and cultural) from 1990–2040. The regulating service contributed the highest ESV in all reference years as per the coefficient of both Costanza et al. (1997) and their own modified (2014). As per the coefficient value of 1997, the ESV of the regulating service was USD 1331.16 × 10³, USD 1409.68 × 10³, USD 1339.43 × 10³, USD 1412.95 × 10³, USD 1384.24 × 10³, and USD 1329.82 × 10³ in the years 1990, 2000, 2010, 2021, 2030, and 2040, respectively, while it was USD 1931.75 × 10³, USD 2035.55 × 10³, USD 1912.44 × 10³, USD 2090.27 × 10³, USD 2081.03 × 10³, and USD 2030.6 × 10³ in the same years as per the coefficient value of 2014. The cultural service has the lowest contribution of ESV in all study years as per both coefficient values (1997 and 2014). It was only USD 268.74 × 10³, USD 291.26 × 10³, USD 244.46 × 10³, USD 321.01 × 10³, USD 326.62 × 10³, and USD 317.13 × 10³ in the years 1990, 2000, 2010, 2021, 2030, and 2040, respectively, while it was USD 752.79 × 10³, USD 832.14 × 10³, USD 848.46 × 10³, USD 896.04 × 10³, USD 878.59 × 10³, and USD 585.57 × 10³ in the same years as per Costanza et al. (2014).

*4.3. Spatio-Temporal Variation of ESV*

The spatio-temporal variation of ESV has been analyzed in response to LULC alteration. Figure 4 and Table 10 show the spatio-temporal variation of ESV based on coefficient values as given by Costanza et al. (1997) from 1990–2040. In 1990, the area under the very high category was 52.18 percent of the total area and was predicted to be 41.36 percent in 2040. Due to the ongoing change in land-use patterns in the study region, the area of this category has steadily declined since 2020. The area under very low, low, moderate, and high was 0.98 percent, 9.52 percent, 17.01 percent, and 20.30 percent in the year 1990, respectively, and would be 0.84 percent, 10.25 percent, 15.49 percent, and 32.05 percent in 2040. The low and high categories increased during the study period (1990–2040), whereas the very low and moderate categories experienced some decline.

**Table 9.** Estimated values for different ecosystem functions in the study area in the period 1990–2040.

| ESV Function | 1990 | | 2000 | | 2010 | | 2021 | | 2030 | | 2040 | |
|---|---|---|---|---|---|---|---|---|---|---|---|---|
| | \*Cc1997 | Cc2014 | Cc1997 | Cc2014 | Cc1997 | Cc2014 | Cc1997 | Cc2014 | Cc1997 | Cc2014 | Cc1997 | Cc2014 |
| *Provisioning* | | | | | | | | | | | | |
| Food production | 37.22 | 560.2 | 36.79 | 584.83 | 35.79 | 530.47 | 36.44 | 646.65 | 36.76 | 713.03 | 37.03 | 767.51 |
| Raw materials | 59.2 | 124.45 | 54.04 | 127 | 55.21 | 114.99 | 47.26 | 133.86 | 41.35 | 137.85 | 36.34 | 138.73 |
| Genetic resource | 6.23 | 329.29 | 5.57 | 329.29 | 5.88 | 310.9 | 4.68 | 342.22 | 3.99 | 361.38 | 3.43 | 377.92 |
| Water supply | 871.08 | 197.63 | 953.33 | 198.94 | 822.99 | 215.55 | 1036.14 | 179.82 | 1051.63 | 176.52 | 1023.75 | 178.96 |
| Total | 973.73 | 1211.57 | 1049.73 | 1240.06 | 919.87 | 1171.91 | 1124.52 | 1302.55 | 1133.73 | 1388.87 | 1100.55 | 1463.12 |
| *Regulating* | | | | | | | | | | | | |
| Gas regulation | 27.45 | 1.13 | 30.56 | 1.01 | 24.22 | 1.06 | 35.06 | 0.85 | 36.33 | 0.72 | 35.58 | 0.62 |
| Disturbance regulation | 750 | 229.24 | 832.99 | 253.55 | 661.58 | 202.54 | 957.35 | 289.98 | 991.92 | 299.74 | 971.37 | 293.14 |
| Waste treatment | 34 | 33.99 | 30.48 | 30.4 | 31.86 | 31.86 | 25.58 | 25.54 | 21.74 | 21.74 | 18.67 | 18.67 |
| Erosion control | 37.44 | 243.8 | 33.44 | 263.8 | 35.05 | 218.03 | 28.09 | 295.31 | 23.91 | 304.57 | 20.56 | 299.51 |
| Pollination | 3.56 | 6.67 | 3.78 | 6.66 | 3.38 | 6.31 | 4.33 | 6.93 | 4.93 | 7.34 | 5.44 | 7.7 |
| Climate regulation | 55.05 | 328.97 | 49.23 | 327.17 | 51.61 | 326.89 | 41.36 | 330.83 | 35.27 | 324.77 | 30.24 | 319.22 |
| Biological control | 6.84 | 125.11 | 7.13 | 128.35 | 6.56 | 113.45 | 7.93 | 134.26 | 8.85 | 133.25 | 9.64 | 127.85 |
| Water regulation | 218.93 | 636.42 | 206.67 | 669.91 | 333.47 | 702.1 | 84.32 | 617.89 | 30.94 | 583.75 | 14.63 | 556.17 |
| Waste treatment | 197.89 | 326.42 | 215.4 | 354.7 | 191.7 | 310.2 | 228.93 | 388.68 | 230.35 | 405.15 | 223.69 | 407.72 |
| Total | 1331.16 | 1931.75 | 1409.68 | 2035.55 | 1339.43 | 1912.44 | 1412.95 | 2090.27 | 1384.24 | 2081.03 | 1329.82 | 2030.6 |
| *Supporting* | | | | | | | | | | | | |
| Nutrient cycling | 141.06 | 147.14 | 126.14 | 159.41 | 132.23 | 130.82 | 105.97 | 178.08 | 90.21 | 181.94 | 77.47 | 176.74 |
| Soil formation | 3.97 | 103.51 | 3.55 | 109.45 | 3.78 | 98.42 | 2.98 | 124.19 | 2.54 | 140.59 | 2.18 | 154.63 |
| Habitat/refugia | 45.45 | 359.41 | 50.42 | 361.3 | 40.04 | 326.73 | 57.96 | 366.8 | 60.07 | 355.74 | 58.82 | 334.81 |
| Total | 190.48 | 610.06 | 180.11 | 630.16 | 176.05 | 555.97 | 166.91 | 669.07 | 152.82 | 678.27 | 138.47 | 666.18 |
| *Cultural* | | | | | | | | | | | | |
| Cultural | 183.08 | 149.64 | 203.21 | 166.19 | 161.55 | 132.02 | 233.38 | 190.97 | 241.77 | 197.85 | 236.66 | 193.34 |
| Recreation | 85.66 | 603.15 | 88.05 | 656.95 | 82.91 | 716.82 | 87.63 | 705.07 | 84.85 | 680.74 | 80.47 | 665.23 |
| Total | 268.74 | 752.79 | 291.26 | 823.14 | 244.46 | 848.46 | 321.01 | 896.04 | 326.62 | 878.59 | 317.13 | 858.57 |
| Total ESV of all function | 2764.11 | 4506.17 | 2930.78 | 4728.91 | 2679.81 | 4489.16 | 3025.39 | 4957.93 | 2997.41 | 5026.67 | 2885.97 | 5018.47 |

Header note: Ecosystem Service Value (in USD 10³)

\*Cc represents Costanza et al. coefficients.

**Table 10.** Spatio-temporal variation of ESV as per Costanza et al. (1997).

| ESV Categories | ESV Range (in USD 10³) | Percentage of Area | | | | | |
|---|---|---|---|---|---|---|---|
| | | 1990 | 2000 | 2010 | 2021 | 2030 | 2040 |
| Very low | Below 12,000 | 0.98 | 0 | 0 | 0 | 0 | 0.84 |
| Low | 12,000–16,000 | 9.52 | 2.53 | 8.50 | 0.18 | 5.86 | 10.25 |
| Moderate | 16,000–20,000 | 17.01 | 14.71 | 18.73 | 20.08 | 17.17 | 15.49 |
| High | 20,000–24,000 | 20.30 | 25.31 | 23.70 | 33.35 | 34.63 | 32.05 |
| Very high | Above 24,000 | 52.18 | 57.44 | 49.08 | 46.38 | 42.33 | 41.36 |

Figure 5 and Table 11 show the spatio-temporal variation of ESV as per the revised coefficient value of Costanza et al. (2014). The ESV was again classified into five categories: very low (<USD 30,000), low (USD 30,000–33,000), moderate (USD 33,000–36,000), high (USD 36,000–39,000), and very high (>USD 39,000). The range of ESV of these categories increased in this table because Costanza et al. (2014) increased the coefficient values from the previous. According to the updated coefficient (2014), the area falling under the very high category has been steadily shrinking during the research period; it was 7.17 percent in 1990 and will be zero in 2040. The very low and high categories also decreased over the whole study period. In 1990, the area under very low and high categories was 14.10 percent and 25.95 percent, respectively, while it would be 9.54 percent and 11.88 percent in 2040.

On the other hand, the low and moderate categories have increased from 22.16 percent and 30.60 percent in 1990 to 30.80 percent and 47.78 percent, respectively, in 2040.

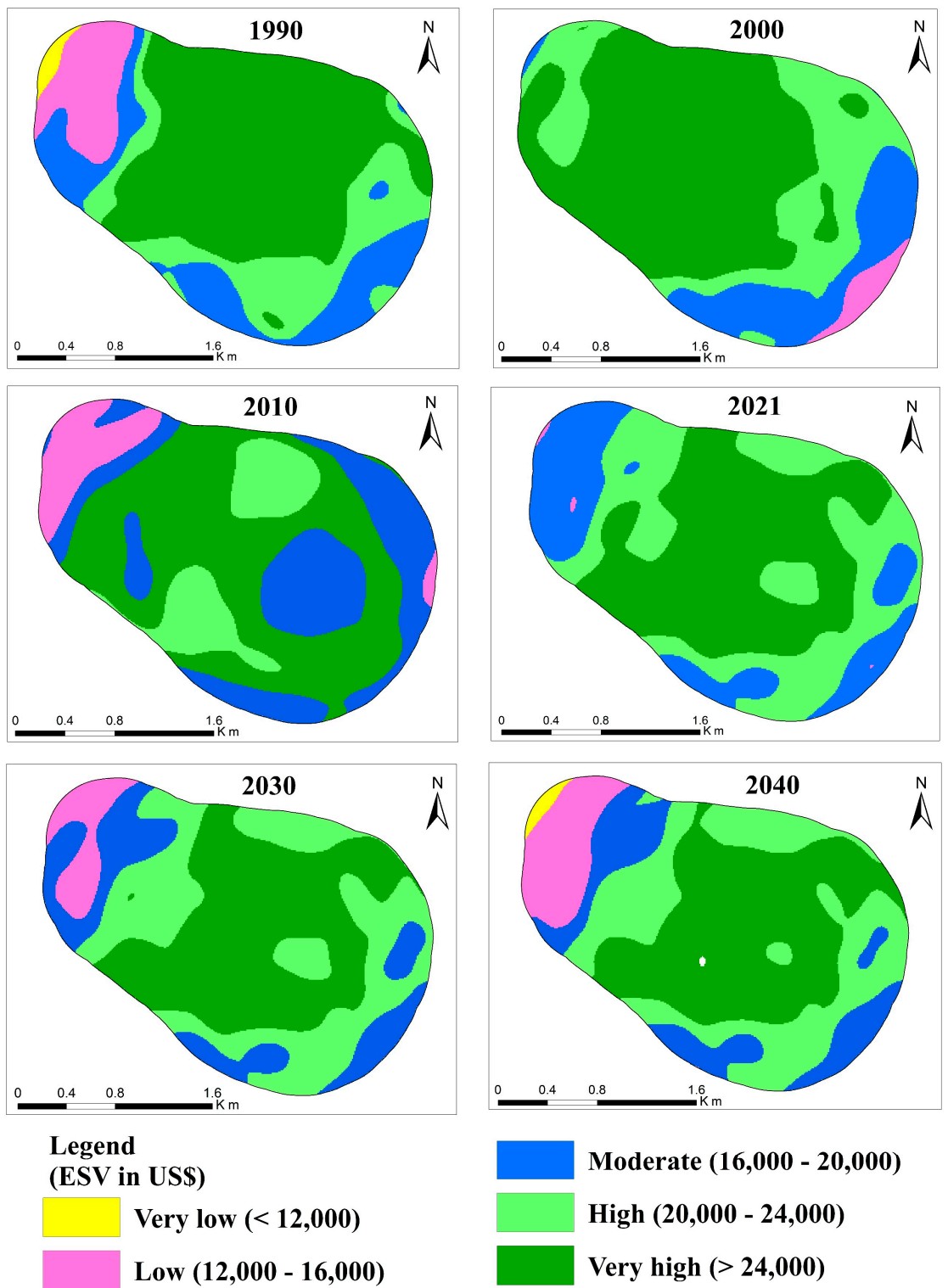

**Figure 4.** Spatio-temporal differences of the ESV (in USD $10^3$) based on Costanza et al. (1997).

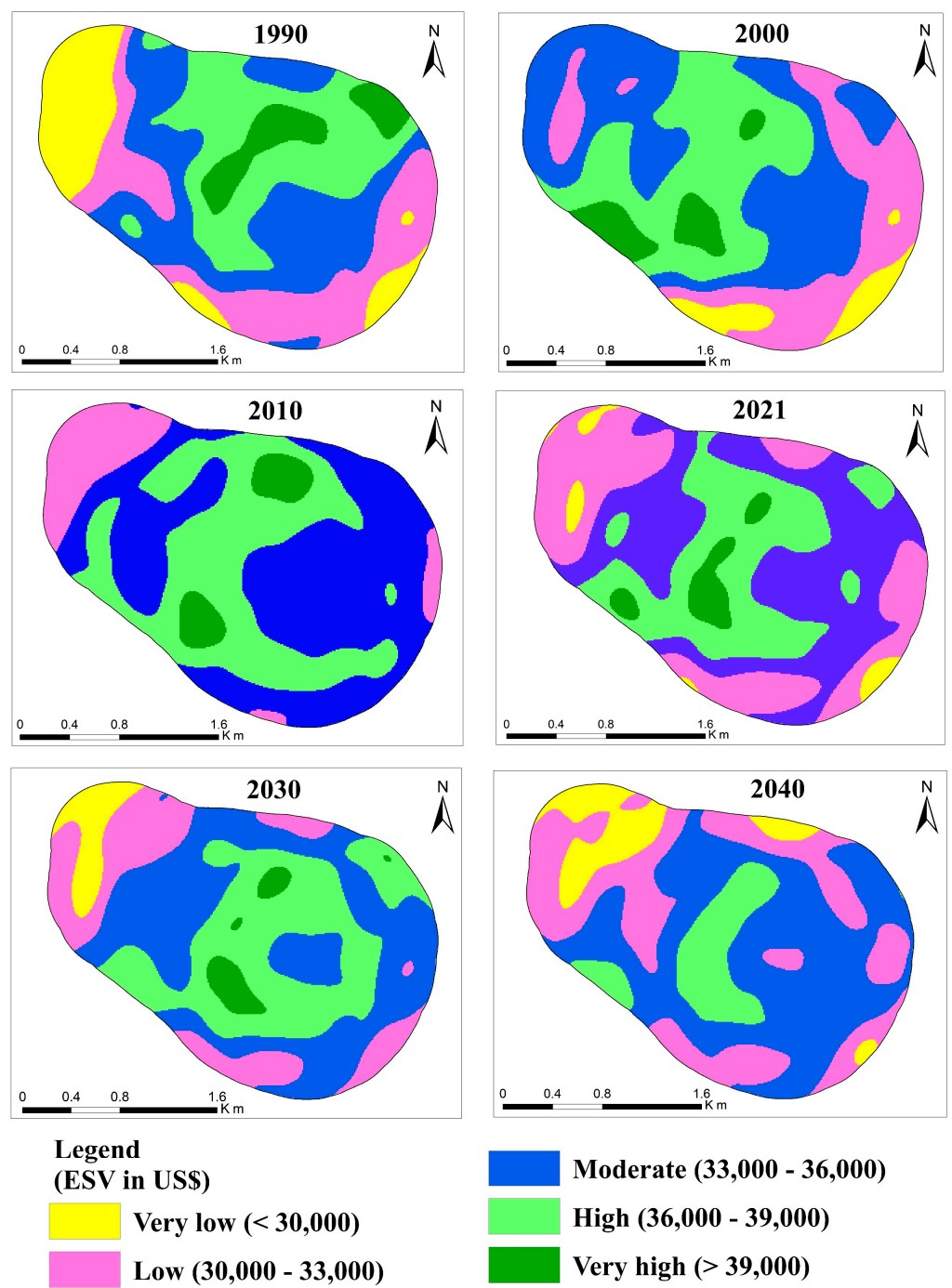

**Figure 5.** Spatio-temporal differences of the ESV (in USD $10^3$) based on Costanza et al. (2014).

**Table 11.** Spatio-temporal variation of ESV as per Costanza et al. (2014).

| ESV Categories | ESV Range (in USD $10^3$) | Percentage of Area | | | | | |
|---|---|---|---|---|---|---|---|
| | | **1990** | **2000** | **2010** | **2021** | **2030** | **2040** |
| Very low | Below 30,000 | 14.1 | 4.96 | 0 | 2.4 | 5.21 | 9.54 |
| Low | 30,000–33,000 | 22.16 | 22.03 | 11.31 | 26.1 | 18.28 | 30.8 |
| Moderate | 33,000–36,000 | 30.6 | 38.65 | 49.73 | 39.77 | 40.74 | 47.78 |
| High | 36,000–39,000 | 25.95 | 28.54 | 34.05 | 27.4 | 32.42 | 11.88 |
| Very high | Above 39,000 | 7.17 | 5.8 | 4.9 | 4.32 | 3.33 | 0 |

The CS is an important parameter used in the estimation of ESV. It is a measure of the sensitivity of the estimated value to changes in the input parameters used in the estimation. The CS is used to determine the level of uncertainty associated with the estimated value of an ecosystem service. The sensitivity of ESV to changes in the value coefficient should be relatively low for a reliable result (CS < 1). Here, the CS was <1 in all land-use types, representing that the total ESV estimated in the study site was relatively inelastic (low sensitive) with respect to the value coefficient (Tables 12 and 13). As per the 1997 coefficient, the CS values for open water and aquatic plants show relatively high sensitivity to changes in LULC, with CS values ranging from 0.01 to 0.91 for aquatic plants and 0.01 to 0.19 for open water. This revealed that the estimated ESV for these LULC types is more sensitive to changes in LULC, and small changes in LULC can have a significant impact on the estimated ESV as compared to other classes. The CS values for vegetation and agricultural land are relatively low, with values ranging from 0.01 to 0.14 for vegetation and 0.01 for agricultural land. The CS value for the built-up class was zero because there was no coefficient value associated with this class. As per the coefficient of 2014, the CS values for open water show a relatively low sensitivity, ranging from 0.08 in 1990 to 0.01 in 2030. This suggests that the estimated ESV for open water is relatively stable and less sensitive. In contrast, the CS values for aquatic plants are relatively high, ranging from 0.3 in 1990 to 0.51 in 2030, revealing that the aquatic plants class is comparatively more sensitive, and small changes in LULC can have a significant impact on the ESV. The CS values for vegetation and agricultural land are moderate, with values ranging from 0.12 to 0.32 for both LULC types. This suggests that the estimated ESV for these LULC types is moderately sensitive. The CS values for built-up areas are relatively low, ranging from 0.02 to 0.06, suggesting that the estimated ESV for built-up areas is relatively stable and less sensitive to changes. The CS was highest for aquatic plants because of the high coefficient value for this particular land-use type. Overall, the sensitivity value indicated that the ESV estimation was strong, despite uncertainties in the value coefficient.

**Table 12.** Coefficient of sensitivity (CS) value of different LULC classes based on Costanza et al.'s (1997) ESV value.

| LULC Types | Coefficient of Sensitivity (CS) | | | | | |
|---|---|---|---|---|---|---|
| | **1990** | **2000** | **2010** | **2021** | **2030** | **2040** |
| Open water | 0.12 | 0.11 | 0.19 | 0.04 | 0.01 | 0.01 |
| Aquatic plant | 0.73 | 0.77 | 0.67 | 0.86 | 0.89 | 0.91 |
| Vegetation | 0.14 | 0.12 | 0.13 | 0.09 | 0.08 | 0.07 |
| Agricultural land | 0.01 | 0.01 | 0.01 | 0.01 | 0.01 | 0.01 |
| Built-up | - | - | - | - | - | - |

**Table 13.** Coefficient of sensitivity (CS) value of different LULC classes based on Costanza et al.'s (2014) ESV value.

| LULC Types | Coefficient of Sensitivity (CS) | | | | | |
|---|---|---|---|---|---|---|
| | **1990** | **2000** | **2010** | **2021** | **2030** | **2040** |
| Open water | 0.08 | 0.07 | 0.12 | 0.03 | 0.01 | 0 |
| Aquatic plant | 0.3 | 0.45 | 0.38 | 0.5 | 0.51 | 0.5 |
| Vegetation | 0.24 | 0.2 | 0.22 | 0.16 | 0.14 | 0.12 |
| Agricultural land | 0.23 | 0.23 | 0.22 | 0.26 | 0.29 | 0.32 |
| Built-up | 0.02 | 0.04 | 0.05 | 0.06 | 0.06 | 0.06 |

## 5. Discussion

### 5.1. Causes and Trends of LULC Dynamics

Previous research has shown that LULC alterations remarkably impact an ecosystem's primary ecological processes, including energy exchange, soil erosion and accumulation,

and the water and biogeochemical cycles, thus affecting ecosystem services [42,45]. The global LULC has increasingly been changing due to rapid population growth, urban area expansion, and agricultural intensification that affect regional ecosystem services [10]. The Brahmaputra Valley of Assam, India, has experienced a significant change in LULC that is influenced by anthropogenic and natural activities [46], which affect the ecosystem services of the wetlands in the valley. The ecosystem services of the Merbil wetland and its surrounding area have been changing due to the continuous alteration of land use/land cover over time. The present study observed a significant expansion of built-up area (205.98 percent), agricultural land (53.15 percent), and aquatic plant (29.61 percent), and reduction of open water (−95.61) and vegetation (−45.08 percent) during 1990–2040 (Table 5). In 1990, vegetation was the largest land-use class in the study site, but it continuously decreased over the study period and was transformed into built-up and agricultural land. The huge migration of people from the nearby region is the primary reason for this gradual loss of vegetation. The majority of previously vegetated lands were removed and altered to agricultural land, leading to a steady increase in agricultural land from 1990 to 2040. The area of open water has decreased in the study period (1990–2040) mainly due to the expansion of aquatic plants. As per the observation, most of the area under open water has been invaded by water hyacinth (*Eichhornia crassipes*), an invasive aquatic weed since 2010 that has accelerated the decline of open water.

### 5.2. Change of ESV in Response to LULC

The LULC alteration is the major cause of ESV loss or the shift of types of ESV from one to another [47,48]. The total ESV provided by the LULC types has been assessed for 1990, 2000, 2010, and 2021 as well as predicted for 2030 and 2040 using the global value coefficient given by Costanza et al. (1997 and 2014). In this study, we observed that the total ecosystem services value increased by 9.45 percent and 9.97 percent as per the coefficient of 1997 and 2014, respectively, during the observed period (1990–2021). The increase of built-up, aquatic plants and agricultural land were the main causes of these changes. Most vegetation-covered land and open water surface have been altered into agricultural land and aquatic plants, respectively, throughout this time. The built-up area does not offer any ecosystem services, as per the coefficient value of 1997, but Costanza et al. changed the coefficient in 2014 and gave a value for the built-up area that is higher than vegetation and agricultural land. Hence, the ESV has increased during this period with the steady growth of the built-up class. During the predicted period (2021–2040), the ESV would be reduced by 4.60 percent according to Costanza et al.'s (1997) coefficient due to a significant decrease in open water and vegetation, while the ESV would be slightly increased (1.22 percent) according to the 2014 coefficient due to growth in the built-up class. Even though the water body areas were quite small, they had a significant impact on the ecosystem service, which was the main driver of improvement for the local ecological environment [49]. However, it must be noted we have not considered price changes in the future. Many open water surface areas would be converted into built-up areas and agricultural land during the projected period, reducing ESV during that time. Although the expansion of agriculture may seem economically advantageous, a significant expansion of agricultural area can lead to the destruction of natural ecosystems. The intricacy of human interaction in the commercial agricultural landscape and its effects on ecosystem services are often ignored by the present agricultural paradigm, which prioritizes crop production over other ecosystem services [50]. The vegetation-covered area decreased by 45.08 percent between 1990 and 2040, as shown in Table 5, which is not a good sign for the study area. Natural vegetation is beneficial to humans and a major source of ecosystem services. Despite their significance, vegetation cover has changed over the past few decades throughout the study site as a result of both direct and indirect causes, such as population pressure, agricultural expansion, fuel-wood collection, and infrastructural development. This depletion of vegetation cover affects ecosystem integrity by decreasing the accessibility of goods and services like carbon regulation, water regulation, and biodiversity due to

species disappearance, habitat loss, and changes in species distribution [51]. Additionally, changes in LULC are connected with other global phenomena, such as climate change, biodiversity loss, and land degradation [49], which directly or indirectly impact regional ecosystem services. A similar study was conducted by Sannigrahi [40] to determine the global value of ecosystem services and revealed that the average value was USD 58.97 trillion per year in 1995 and USD 57.76 trillion per year in 2015. This indicates a net loss of ESV, amounting to USD 1.21 trillion per year, during the analysis period from 1995 to 2015. This loss can be attributed to the reduction in forest cover and wetland and water surface area. Fenqine et al. [52] conducted a study on the wetland loss history and its impact on ecosystem services in the Sanjiang Plain of northern China. The study found that since the mid-1950s, the wetland area has declined by 73.3% (about 2.77 million hectares) due to wetlands being transformed into farmland. This has resulted in a reduction of ESV by USD 57.46 billion over the past six decades. The largest reduction in ESV was due to agricultural expansion, particularly that of dry farmland. With the exception of agricultural product functions, all ecosystem services functions have decreased over the past 60 years. Long [45] estimated the changes in ESV associated with LULC changes in Dongting Lake from 1995 to 2020. They observed that the wetland ESV showed a trend of first decreasing and then increasing, resulting in a total loss of ¥87,000.99 million in wetland ESV.

The ESV of the study site was classified into five categories based on grid-wise analysis to study the spatial and temporal distribution. These categories were very low, low, moderate, high, and very high. The very low ESV category was primarily found in the interlacing zone of vegetation and paddy fields, where human activity harmed the natural environment more seriously. The area has sparse vegetation cover and limited agricultural production. The low and moderate classes were found in areas of dense vegetation cover and agricultural land. The high and very high ESV categories were primarily found in areas covered by open water and aquatic plants.

### 5.3. Comparison of ESV Changes between the Value Coefficients Applied

Results from earlier studies demonstrate that the distribution pattern of LULC type in a region, as well as the coefficient value used, have an impact on the pattern of ESV [23,26]. In this study, Costanza et al.'s two value coefficients in 1997 and 2014 were applied, and the changes in ESV show significant variance between these two coefficient values. In the 1997 value coefficient of Costanza et al., the agricultural land had a low coefficient value, while the built-up area had none. Conversely, the agricultural land received a high value per the value coefficient of 2014, and concurrently, the coefficient value was added for the built-up class. Hence, there occurs a variation in the results based on the two value coefficients. This is due to the fact that different LULC types experience different rates of land-use changes over time, and the LULC change is a significant element of variation in ESV alteration.

### 5.4. Limitations of This Study

The study has several limitations that need to be considered when interpreting the results. One significant limitation is the resolution of satellite images and the algorithms used for LULC classification, which can introduce errors in the classification process which may affect the accuracy of the analysis. Additionally, due to the limitations of satellite images, LULC was not classified with a high level of detail, which may impact the study's outcomes [53–57].

Another limitation concerns the CA-Markov model, which, while effective in simulating LULC patterns, may not fully account for human disturbances and spatial changes in the landscape scenario. This can lead to uncertainties and errors in the simulation results, stemming from limitations in image classification techniques and uncertainties in the transition rules within the model.

Furthermore, the study relies on the global coefficient values provided by Costanza et al. for estimating and predicting ESVs. However, these global coefficients may not always precisely estimate the economic worth at a local level due to varying economic conditions.

For instance, the coefficients assign higher values to urban areas compared to forests and agriculture, which could be detrimental to natural ecosystems. Additionally, the LULC types in this study are not completely similar to the biomes used by Costanza et al., which may impact the accuracy of ESV estimation in the research area.

Moreover, the coefficient method employed in this study may not be suitable for valuing all types of ecosystem services, particularly those that do not have a direct market value or are not easily quantifiable. This limitation further underscores the challenges in accurately estimating the value of ecosystem services in various contexts [58].

Despite these limitations, the present study provides valuable insights into the LULC changes needed to achieve sustainability goals for ecosystem services and minimize future ESV risk. It also informs policy implementation for environmental and ecological enhancement in the area [59,60]. However, future research should consider addressing these limitations by refining the methods used for LULC classification, improving the CA-Markov model, and exploring alternative valuation methods to better estimate ESV at local and national levels.

## 6. Conclusions

In conclusion, valuing ecosystem services in monetary terms serves as a valuable tool for raising public awareness about the scarcity of natural resources and the benefits they provide, many of which are not traded on the market. This study examined the dynamics of ESV in response to LULC changes, assessing past and predicting future LULC and ESV trends in the study area from 1990 to 2040. The results revealed significant LULC changes, with notable increases in built-up areas, agricultural land, and aquatic plants, and decreases in open water and vegetation-covered areas. These changes have considerably impacted the study area's ESV, emphasizing the need to safeguard vegetation and natural water bodies while promoting sustainable land use and development practices to preserve the regional ecological structure.

However, several limitations were encountered in this study, including the resolution of satellite images, algorithm limitations in LULC classification, and the CA-Markov model's inability to fully account for human disturbances and spatial changes in the landscape scenario. Moreover, the global coefficient values provided by Costanza et al. may not accurately estimate the ESV of the specific LULC types in our study area, and the coefficient method may not be suitable for valuing all types of ecosystem services, particularly those without a direct market value or that are not easily quantifiable.

Despite these limitations, the study offers valuable insights into the LULC changes required to achieve sustainability goals for ecosystem services and minimize future ESV risks. It also contributes to policy implementation to enhance the environment and ecology in the area.

For future research, the authors recommend using similar methods at the national level, as studies on ESV change have been conducted in many countries at local, regional, and national levels. However, there have only been a few small-scale studies conducted in India, emphasizing the need for such investigations in India at both local and national levels. Efforts should also be made to address the limitations encountered in this study, refining LULC classification methods, improving the CA-Markov model, and exploring alternative valuation methods to enhance the accuracy of ESV estimation in future research.

**Author Contributions:** Durlov Lahon, Dhrubajyoti Sahariah, Jatan Debnath, Gowhar Meraj, Shizuka Hashimoto: conceptualization, methodology, software, data curation, and writing—original draft preparation. Durlov Lahon, Jatan Debnath, Gowhar Meraj: writing—review and editing. Nityaranjan Nath, Majid Farooq, Pankaj Kumar, Shizuka Hashimoto: conceptualization, methodology, writing—review, and editing. All authors have read and agreed to the published version of the manuscript.

**Funding:** The first author Durlov Lahonis grateful to the University Grants Commission, New Delhi, for the grant providing him with a Junior Research Fellowship. The author Jatan Debnath

**Institutional Review Board Statement:** Not applicable.

**Informed Consent Statement:** Not applicable.

**Data Availability Statement:** The data will be available at a reasonable request from the corresponding author.

**Acknowledgments:** The authors cordially acknowledge the GSI (Geological Survey of India) for providing toposheets, and USGS (United State Geological Survey) for providing the satellite images at no cost. The authors are also grateful to the Department of Geography, Gauhati University, Assam, India. Moreover, they would like to extend thanks to the local people for their enthusiastic cooperation during field verification.

**Conflicts of Interest:** The authors declare no conflict of interest.

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
