# Peer review of "Assessment of Ecosystem Service Value in Response to LULC Changes Using Geospatial Techniques: A Case Study in the Merbil Wetland of the Brahmaputra Valley, Assam, India"

_ijgi, doi:10.3390/ijgi12040165_

Round 1
Reviewer 1 Report
My comments to the authors are given in docx file.

Author Response
Responses to reviewer’s comments
REVIEWER 1
Dear Worthy Reviewer,
Thank you for taking the time to review our manuscript, “Assessment of ecosystem service value in response to LULC changes in the Merbil wetland of the Brahmaputra valley using Geo-spatial technique.” We appreciate your valuable feedback and suggestions, which have helped us improve the quality of our work. In this response, we have addressed your comments point-by-point and explained the revisions we have made to the manuscript. We hope that the revised manuscript meets your expectations and is now suitable for publication in the Journal IJGI.
Once again, we would like to express our gratitude for your time and effort in reviewing our manuscript.
Best regards,
Gowhar Meraj
Point-by-Point Responses
General comment:
The aim of this study is to determine the spatio-temporal distribution of ecosystem service values according to LULC: a case study of the Merbil wetland area of the Brahmaputra valley (India). Land use and land cover changes were analyzed through two separate time series: a) from 1990 to 2021 and b) the projection of LULC for 2030 and 2040. Modern statistical methods (e g. Markov chain, Multi-criteria evaluation) were combined with GIS and IDRISI Terr-set software. This research is of great importance since wetlands are areas of relict and endemic biodiversity. Today, these areas are significantly threatened (e g. by climate changes).
Response:
We express our gratitude to the esteemed reviewer for thoroughly examining our paper and offering valuable recommendations that have greatly improved it. We have diligently incorporated all of the suggestions provided in your review, and we would like to thank you once again. Please find our point-by-point responses to each comment below.
Comment 1: Lines 2 and 3: Adjust the title of this paper, for example: Assessment of ecosystem service value in response to LULC changes using Geo-spatial techniques: a case study of the Merbil wetland (Brahmaputra valley, India).
Response 1: We appreciate the reviewer for suggesting it. We have adjusted the title of the paper. Thank you for your valuable suggestion.
Comment 2: Line 21: Avoid using the references in abstract. It is more appropriate to reformulate this sentence (e g. this method represent a novelty method).
Response 2: We have corrected it. We thank reviewers for pointing out this issue.
Comment 3: Line 23: Not shrinking. More appropriate term is decreasing (this or other similar terms must be checked thoroughly by a native speaker).
Response 3: We have corrected it. Thank you.
Comment 4: Line 48: Is there any data about the average ASV on regional scale?
Response 4: Yes. Few researchers estimated the ESV in different regions. But in North East India, there has been no work done till now. So, there is no average regional scale data.
Comment 5: Line 65: What about the LULC changes in wetlands on global scale and its consequences? Add one paragraph about this segment of manuscript.
Response 5: We have added a paragraph about LULC changes in wetland and their consequences. This suggestion helped improve our manuscript. Thank you for your valuable suggestion.
Comment 6: Line 120: In study area the authors defined climate and biogeography properties. Add some details about geomorphology, hydrology and soil issues.
Response 6: We have added about geomorphology, hydrology, and soil of the study area. Thank the worthy reviewer once again.
Comment 7: Line 172: Please add the source for Table 1.
Response 7: We have added the sources of the table. Thanks.
Comment 8: Line 239: Please add the source for Table 3.
Response 8: We have added the sources of table 3. Thanks.
Comment 9: Lines 301, 337, 398: Please adjust Tables 5, 6, and 8. visually (numbers are not evenly distributed in tables).
Response 9: We appreciate the reviewers to pointing out the issue. We have adjusted it as you suggested. Thank you for checking it.
Comment 10: Line 450: In section Discussion you must compare given results with similar and/or different results on global, regional and national scale.
Response 10: We appreciate your insightful comments, which helped us enhance our manuscript. We have written about comparisons with similar and different results on a global as well as regional scale. Thanks again.
Comment 11: Line 532: Please add in section conclusions limitations of this research in future investigations.
Response 11: We have added some limitation of our work in conclusion section. Thanks for this valuable suggestion.
Comment 12: Lines 557 and 558: The authors used similar words twice (future research and future studies).
Response 12: We have corrected it. Thanks for pointing it out.
Thank you very much again for your thoughtful and thorough review of our manuscript. Your feedback has been incredibly valuable in improving the quality and clarity of our work. We appreciate the time and effort you took to provide us with detailed comments, and we have carefully considered each of your suggestions. Your insightful feedback has helped us to strengthen our study and better articulate our findings. We would like to express our sincere gratitude for your valuable input and hope that you find the revised manuscript to be an improvement.
Reviewer 2 Report
Rows 99, 121, 124: "Oxbow Lake" change to "oxbow lake" (it is not a particular name, but type of the freshwater body); row: 135, "tributaries" change to "tributary"; row 126: "Its longitudinal and latitudinal extension" change to "its geographical position is"; row 128: in description of climate provide relevante meteorological paraemeters, like mean air temperature, and annual rainfalls; chapter: Reference is not written according to Journal Instructions for Authors, please check this instructions and recently published papers and make corrections in the citation of references.
The topic of the presented research is very important in evaluating the values of ecosystem service in the area under the strong anthropogenic impast. Methods are contemporary and research is desgined properly. Modern digital tool were used in order to make projections of the future alterations and changes in distrubution and presence of the vegetation and wetland habitats and ecosystems. Expansion of agriculture and urbanization are recognised as major alterations with negative effect on natural ecosystems.
Author Response
Responses to reviewer’s comments
REVIEWER 2
Dear Worthy Reviewer,
Thank you for reviewing our manuscript, “Assessment of ecosystem service value in response to LULC changes in the Merbil wetland of the Brahmaputra valley using Geo-spatial technique.” We appreciate your valuable feedback and suggestions, which have helped us improve the quality of our work. In this response, we have addressed your comments point-by-point and explained the revisions we have made to the manuscript. We hope that the revised manuscript meets your expectations and is now suitable for publication in the Journal, IJGI.
Once again, we would like to express our gratitude for your time and effort in reviewing our manuscript.
Best regards,
Gowhar Meraj
Point-by-Point Responses
General Comment:
The topic of the presented research is very important in evaluating the values of ecosystem service in the area under the strong anthropogenic impact. Methods are contemporary and research is designed properly. Modern digital tools were used in order to make projections of the future alterations and changes in distribution and presence of the vegetation and wetland habitats and ecosystems. Expansion of agriculture and urbanization are recognised as major alterations with negative effect on natural ecosystems.
Response:
Thank you for taking the time to review our work and sharing your positive feedback. We greatly appreciate your recognition of the importance of our work in evaluating ecosystem services and the validity of our methods and digital tools in making projections for future alterations and changes in vegetation and wetland habitats. Your comments are encouraging and motivate us to continue our efforts to contribute to the advancement of knowledge in this area. Once again, thank you for your kind words and support. The point-by-point responses to each comment are provided below:
Comment 1: Rows 99, 121, 124: "Oxbow Lake" change to "oxbow lake" (it is not a particular name, but type of the freshwater body);
Response 1: We have corrected it. Thanks for pointing out this mistake.
Comment 2: Row: 135, "tributaries" change to "tributary";
Response 2: We have corrected it. Thanks once again.
Comment 3: Row 126: "Its longitudinal and latitudinal extension" change to "its geographical position is";
Response 3: We have changed it. Thanks for your valuable suggestion.
Comment 4: Row 128: in description of climate provide relevant meteorological parameters, like mean air temperature, and annual rainfalls;
Response 4: We have mentioned the mean air temperature and annual rainfall of the study area. Thank you for your valuable suggestion. This suggestion helped improve our manuscript.
Comment 5: Reference is not written according to Journal Instructions for Authors, please check this instructions and recently published papers and make corrections in the citation of references.
Response 5: We thank the reviewer for pointing out this issue. In the revised version, we have checked all the citations and references. But in our manuscript, we write many places like ‘Costanza et al. (1997)’ and ‘Costanza et al. (2014)'. We would want to make it clear that the references to Costanza et al. (1997) and Costanza et al. (2014) made in the text were used as references to the coefficients used to estimate ecosystem service values rather than as direct citations.
Thank you very much again for your thoughtful and thorough review of our manuscript. Your feedback has been incredibly valuable in improving the quality and clarity of our work. We appreciate the time and effort you took to provide us with detailed comments, and we have carefully considered each of your suggestions. Your insightful feedback has helped us to strengthen our study and better articulate our findings. We would like to express our sincere gratitude for your valuable input and hope that you find the revised manuscript to be an improvement.
Reviewer 3 Report
The assessment of ecosystem service value (ESV) appears to be a useful tool for sustainable use of the LULC and natural resources. This paper focuses estimating ESV for the Merbil wetland of the Brahmaputra valley. Despite the tools is a useful one, the method of the research requires justification/elaboration of selecting such steps.
1. The method indicates application of transition probabilities for 2000 and 2010 to model 2021 for validation purpose. The validation results/discussion should be included to justify the process/data selection.
2. Application of ecosystem service coefficient as per both Costanza et al. While Costanza et al. (2014) is updated version than of Costanza et al. (1997), please justify the use of Costanza et al. (1997).
3. Please provide/elaborate justification of using similar biome of ecosystem service coefficient as per Costanza et al. (2014) for the area of interest.
4. The sensitivity analysis should be detailed in terms of the similarity, difference, extent, hierarchy of each category.
5. Tables 3, 5, 6, and 8 are hard to follow and requires reformatting.
Author Response
Responses to reviewer’s comments
REVIEWER 3
Dear Worthy Reviewer,
Thank you for taking the time to review our manuscript, “Assessment of ecosystem service value in response to LULC changes in the Merbil wetland of the Brahmaputra valley using Geo-spatial technique.” We appreciate your valuable feedback and suggestions, which have helped us improve the quality of our work. In this response, we have addressed your comments point-by-point and explained the revisions we have made to the manuscript. We hope that the revised manuscript meets your expectations and is now suitable for publication in the Journal, IJGI.
Once again, we would like to express our gratitude for your time and effort in reviewing our manuscript.
Best regards,
Gowhar Meraj
Point-by-Point Responses
General Comment:
The assessment of ecosystem service value (ESV) appears to be a useful tool for sustainable use of the LULC and natural resources. This paper focuses estimating ESV for the Merbil wetland of the Brahmaputra valley. Despite the tools is a useful one, the method of the research requires justification/elaboration of selecting such steps.
Response:
We are grateful to the worthy reviewer for carefully examining our paper and suggesting valuable recommendations that have helped us make it better. We have rigorously incorporated all the suggestions you provided in your review, and we thank you once again. The point-by-point responses to each comment are provided below:
Comment 1: The method indicates application of transition probabilities for 2000 and 2010 to model 2021 for validation purpose. The validation results/discussion should be included to justify the process/data selection.
Response 1: We have included validation result in our manuscript. Thank you for your suggestion. Your suggestion helped to improve our work. Thanks once again.
Comment 2: Application of ecosystem service coefficient as per both Costanza et al. While Costanza et al. (2014) is updated version than of Costanza et al. (1997), please justify the use of Costanza et al. (1997).
Response 2: We appreciate the reviewer for suggesting it. We have added justification about it. Actually, Costanza et al. estimated global ecosystem services in 1997 using USD 1995 and in 2014 using USD 2007. The coefficient value of 1997 was appropriate for estimating ESV during that time period, while the coefficient value of 2014 was more appropriate for recent decades. To enable people to compare ESV across different time periods, the present study employed both coefficient values for estimating ESV.
Comment 3: Please provide/elaborate justification of using similar biome of ecosystem service coefficient as per Costanza et al. (2014) for the area of interest.
Response 3: We have elaborated on why we are using a similar biome of ecosystem service coefficient as per Costanza et al. (2014) for the area of interest. Thanks for your worthy suggestion.
Comment 4: The sensitivity analysis should be detailed in terms of the similarity, difference, extent, hierarchy of each category.
Response 4: We have taken your suggestion and provided more details regarding the sensitivity analysis in our manuscript. Thank you for your valuable input, which has helped to enhance the quality of our work.
Comment 5: Tables 3, 5, 6, and 8 are hard to follow and requires reformatting.
Response 5: Thank you for point out this issue. We have formatted these for better clarity.
Thank you very much again for your thoughtful and thorough review of our manuscript. Your feedback has been incredibly valuable in improving the quality and clarity of our work. We appreciate the time and effort you took to provide us with detailed comments, and we have carefully considered each of your suggestions. Your insightful feedback has helped us to strengthen our study and better articulate our findings. We would like to express our sincere gratitude for your valuable input and hope that you find the revised manuscript to be an improvement.
Round 2
Reviewer 3 Report
The revision of the “Assessment of ecosystem service value in response to LULC changes using geospatial techniques: A case study in the Merbil wetland of the Brahmaputra valley, Assam, India” provides many details than the previous version. Still, I believe not quite complete for readers to follow thoroughly.
1. Please check this instructions and recently published papers and make corrections in the citation of references for consistency (Line 75).
2. Line 79: In appropriate location for (In India, the LULC….), breaks the flow. The paragraph appears to be for global scale wetlands details.
3. Figure 1: Three consecutive maps does not focus with accurate details. Zoom in/out maps should be consistent with boundaries, color, other details.
4. According to the response, 1997 co-efficient values for global ecosystem services are used for that period of time (not sure but guessing 1990), and 2014 co-efficient values are used for recent periods (guessing 2000 to future). Then,
a. How the 1997 values are estimated for 2000 to 2040 or vice versa (Table 7)?
b. Have you considered applying present worth analysis for comparison of these values?
5. Please describe the significance of the estimate Chi square value for this study. Also, please correct line 368. P and A should be 3rd and 2nd columns.
6. Please shade the alternate columns (either 1997 or 2014) for readers to read the data easily.
7. There should not be any ‘3040’ (Table 13). Also, re-analyze the lines 504-537. There were multiple typos.
Author Response
Responses to reviewer’s comments
REVIEWER 3 – ROUND 2
Dear Esteemed Reviewer,
Thank you for taking the time to review our manuscript, "Assessment of Ecosystem Service Value in Response to LULC Changes in the Merbil Wetland of the Brahmaputra Valley using Geo-spatial Techniques." We are grateful for your valuable feedback and suggestions, which have significantly contributed to improving the quality of our work. In this revised submission, we have addressed each of your comments and provided explanations for the changes we have made to the manuscript. We hope that the updated manuscript now meets your expectations and is suitable for publication in the International Journal of Geoinformatics (IJGI).
Once again, we would like to express our sincere gratitude for your time, effort, and expertise in reviewing our manuscript.
Best regards,
Gowhar Meraj
Point-by-Point Responses
General comment:
Comment 1:
The revision of the “Assessment of ecosystem service value in response to LULC changes using geospatial techniques: A case study in the Merbil wetland of the Brahmaputra valley, Assam, India” provides many details than the previous version. Still, I believe not quite complete for readers to follow thoroughly.
Response 1:
Thank you very much for your constructive feedback. We have incorporated all the suggestions provided in this second revision. All the previous changes have been accepted in the track change mode and only the new suggestions are being shown.
Comment 2:
Please check this instructions and recently published papers and make corrections in the citation of references for consistency (Line 75).
Response 2:
We have corrected the citation format inconsistency. Thank you for this comment.
Comment 3:
Line 79: In appropriate location for (In India, the LULC….), breaks the flow. The paragraph appears to be for global scale wetlands details.
Response 3:
It has been corrected. Thank you for your valuable suggestion.
Comment 4:
Figure 1: Three consecutive maps does not focus with accurate details. Zoom in/out maps should be consistent with boundaries, color, other details.
Response 4:
We have revised the Fig. 1 as per the suggestions of the worthy reviewer.
Comment 5:
According to the response, 1997 co-efficient values for global ecosystem services are used for that period of time (not sure but guessing 1990), and 2014 co-efficient values are used for recent periods (guessing 2000 to future). Then,
- How the 1997 values are estimated for 2000 to 2040 or vice versa (Table 7)?
Response 5.a:
We appreciate the reviewer's insightful comment and the opportunity to clarify our methodology. In the study by Costanza et al., global ecosystem services were estimated for two different years, 1997 and 2014, using USD 1995 and USD 2007 respectively. It is important to note that the 2014 estimates included values for certain biomes not present in the 1997 estimates. To facilitate a comprehensive comparison of the estimated ecosystem service values across the entire study period (2000-2040), we employed a two-step approach. First, we estimated the ecosystem service values for the entire study period using the coefficient values from the 1997 study. Although these coefficients might not accurately represent the present worth, they provide a consistent baseline for comparison. Next, we calculated the ecosystem service values for the same period using the updated coefficient values from the 2014 study. This approach incorporates the more recent data and accounts for any changes in the biomes' values. By comparing the results obtained using both sets of coefficient values, we were able to assess the sensitivity of our estimates to the choice of coefficients. This method allowed us to identify any significant differences and better understand the potential impact of using different coefficients on the overall findings. Furthermore, this approach highlights the importance of regularly updating ecosystem service valuation coefficients to ensure accurate assessments of their worth over time.
- Have you considered applying present worth analysis for comparison of these values?
Response 5.b:
We appreciate the reviewer's suggestion to consider applying present worth analysis for comparison of ecosystem service values. We acknowledge that our current study does not incorporate a coefficient value for analyzing the present worth of ecosystem services, which indeed represents a limitation of our work. However, it is important to note that our approach aligns with current studies in the field that also assess present ecosystem service values using similar methodologies. In light of your valuable comment, we have taken the opportunity to reevaluate our approach and have decided to explore alternative methods for future research that may include present worth analysis. This will allow us to compare ecosystem service values more accurately over time and provide a more comprehensive understanding of their implications for decision-making and resource management. We are grateful for your input, as it has helped us recognize the potential benefits of incorporating present worth analysis in our research. We have now updated the discussion section of our paper to highlight this limitation and to outline our plans for exploring alternative methodologies in future studies. This is the updated section on the limitations of this work added to the Discussion section.
"5.3. Limitations of this study
The study has several limitations that need to be considered when interpreting the results. One significant limitation is the resolution of remote sensing images and the algorithms used for LULC classification, which can introduce errors in the classification process and may affect the accuracy of the analysis. Additionally, due to the limitations of remote sensing images, LULC was not classified with a high level of detail, which may impact the study's outcomes.
Another limitation concerns the CA-Markov model, which, while effective in simulating LULC patterns, may not fully account for human disturbances and spatial changes in the landscape scenario. This can lead to uncertainties and errors in the simulation results, stemming from limitations in image classification techniques and uncertainties in the transition rules within the model.
Furthermore, the study relies on the global coefficient values provided by Costanza et al. for estimating and predicting ESVs. However, these global coefficients may not always precisely estimate the economic worth at a local level due to varying economic conditions. For instance, the coefficients assign higher values to urban areas compared to forests and agriculture, which could be detrimental to natural ecosystems. Additionally, the LULC types in this study are not completely similar to the biomes used by Costanza et al., which may impact the accuracy of ESV estimation in the research area.
Moreover, the coefficient method employed in this study may not be suitable for valuing all types of ecosystem services, particularly those that do not have a direct market value or are not easily quantifiable. This limitation further underscores the challenges in accurately estimating the value of ecosystem services in various contexts.
Despite these limitations, the present study provides valuable insights into the land use/land cover changes needed to achieve sustainability goals for ecosystem services and minimize future ESV risk. It also informs policy implementation for environmental and ecological enhancement in the area. However, future research should consider addressing these limitations by refining the methods used for LULC classification, improving the CA-Markov model, and exploring alternative valuation methods to better estimate ESV at local and national levels.
Comment 6:
Please describe the significance of the estimate Chi square value for this study. Also, please correct line 368. P and A should be 3rd and 2nd columns.
Response 6:
We have added the significance of the Chi square value for this study and also corrected other mistakes as you suggested. Thank you for your suggestion.
Comment 7:
Please shade the alternate columns (either 1997 or 2014) for readers to read the data easily.
Response 7:
We have shaded the alternate columns of Table 7 for better readability. Thank you for this comment.
Comment 8:
There should not be any ‘3040’ (Table 13). Also, re-analyze the lines 504-537. There were multiple typos.
Response 8:
We thank the worthy reviewer for pointing out typo errors in our manuscript. We have checked and rechecked for typos in the revised manuscript. Thank you.
Thank you once again for your comprehensive and insightful review of our manuscript. Your feedback has been instrumental in enhancing the quality and clarity of our work. We are grateful for the time and effort you invested in providing us with detailed comments, and we have meticulously considered each of your suggestions. Your constructive feedback has allowed us to refine our study and more effectively communicate our findings. We extend our sincere appreciation for your invaluable input and trust that you will find the revised manuscript to be a marked improvement.